# The Genetic Landscape of Mitochondrial Diseases in Spain: A Nationwide Call

**DOI:** 10.3390/genes12101590

**Published:** 2021-10-09

**Authors:** Marcello Bellusci, Abraham J Paredes-Fuentes, Eduardo Ruiz-Pesini, Beatriz Gómez, Miguel A Martín, Julio Montoya, Rafael Artuch

**Affiliations:** 1Reference Centre for Inherited Metabolic Disorders, 12 de Octubre University Hospital, 28041 Madrid, Spain; marcello.bellusci@salud.madrid.org; 2Biomedical Network Research Centre on Rare Diseases (CIBERER), Instituto de Salud Carlos III, 28029 Madrid, Spain; eduruiz@unizar.es (E.R.-P.); bgomez@ciberer.es (B.G.); 3Clinical Biochemistry Department, Institut de Recerca Sant Joan de Déu, Hospital Sant Joan de Déu, Esplugues de Llobregat, 08950 Barcelona, Spain; abrahamjose.paredes@sjd.es; 4Department of Biochemistry and Molecular Biology, Institute for Health Research of Aragón (IISAragón), University of Zaragoza, 50009 Zaragoza, Spain; 5Mitochondrial & Neuromuscular Disorders Laboratory, Instituto de Investigación Sanitaria 12 de Octubre (imas12), 28041 Madrid, Spain

**Keywords:** mitochondrial diseases, mitochondrial DNA mutations, nuclear DNA mutations, epidemiological data, incidence, Spanish registry

## Abstract

The frequency of mitochondrial diseases (MD) has been scarcely documented, and only a few studies have reported data in certain specific geographical areas. In this study, we arranged a nationwide call in Spain to obtain a global estimate of the number of cases. A total of 3274 cases from 49 Spanish provinces were reported by 39 centres. Excluding duplicated and unsolved cases, 2761 patients harbouring pathogenic mutations in 140 genes were recruited between 1990 and 2020. A total of 508 patients exhibited mutations in nuclear DNA genes (75% paediatric patients) and 1105 in mitochondrial DNA genes (33% paediatric patients). A further 1148 cases harboured mutations in the *MT-RNR1* gene (56% paediatric patients). The number of reported cases secondary to nuclear DNA mutations increased in 2014, owing to the implementation of next-generation sequencing technologies. Between 2014 and 2020, excepting *MT-RNR1* cases, the incidence was 6.34 (95% CI: 5.71–6.97) cases per million inhabitants at the paediatric age and 1.36 (95% CI: 1.22–1.50) for adults. In conclusion, this is the first study to report nationwide epidemiological data for MD in Spain. The lack of identification of a remarkable number of mitochondrial genes necessitates the systematic application of high-throughput technologies in the routine diagnosis of MD.

## 1. Introduction

Mitochondrial diseases (MD) are caused by mutations in nuclear DNA (nDNA) or mitochondrial DNA (mtDNA) genes. Although the paediatric population is probably the most affected, the age of onset varies among patients, and occurrence in adults is relatively frequent. All types of inheritance are expected (sporadic, maternal, X-linked, and autosomal dominant and recessive) [1,2,3].

The general consensus is that MD frequency is greater than that previously expected, and the application of next-generation sequencing (NGS) techniques for diagnostic purposes has substantially increased the diagnostic yield, especially for MD caused by nDNA mutations. The description of new genes has also significantly increased [1,2,3]. There are some previous reports about MD prevalence in specific geographic areas of different countries [4,5,6,7,8,9,10,11,12,13]. Overall, a prevalence of 5–15 cases per 100,000 individuals was estimated [2]. A prevalence of 1 in 4300 for nDNA and mtDNA mutations has been reported for adult populations in the north east of England [6]. In a population-based study of children from western Sweden, the incidence of mitochondrial encephalomyopathies was 1 out of 11,000 in preschool children, and the point prevalence in children under 16 years of age was 1 out of 21,000 [7]. In Spain, the incidence and prevalence of MD have been reported in paediatric and adult populations in certain geographical areas, but the number of cases were limited. The prevalence reported was 8 in 100,000 in the paediatric population and 5.7 in 100,000 in the general adult population in north-west Spain [9]. Recently, an interesting and extensive epidemiological study of MD patients in Japan was reported [13] in which the authors were able to recruit 3629 patients, estimating a prevalence of 2.9 per 100,000. Finally, other studies have reported the prevalence of some specific MD, such as mitochondrial encephalomyopathy with lactic acidosis and stroke-like episodes (MELAS): 0.18 per 100,000 in the general population in Japan [12], or 18.4 in 100,000 in the paediatric population in a region of Finland [11].

With this background, and as a first step for the development of a national clinical registry of MD patients in Spain, our aim was to collect all cases diagnosed during the period from 1990 to 2020 in hospitals and reference centres nationwide. Only patients harbouring either mtDNA or nDNA pathogenic mutations were considered.

## 2. Materials and Methods

### 2.1. Data Source and Patient Selection Protocol

This initiative was the seed for the consolidation of a nationwide Spanish MD registry supervised and controlled by the Biomedical Network Research Centre on Rare Diseases (CIBERER) (acronym GenRaRe, REDCap platform). The variables included in the registry are presented in Appendix A. In order to facilitate data communication and collaboration with other MD registries in the future, the Spanish MD registry was designed following the recommendations of the Italian Registry for MD (Mitocon: https://www.mitocon.it/registro-pazienti/, accessed on 6 September 2021). First, with the aim of describing the genetic landscape of MD in Spain, we called upon all the Spanish clinical units and diagnostic laboratories working in MD diagnosis to report for each case the following variables: affected gene, mode of inheritance, pathological variants reported as HGVS format, year and age of diagnosis (paediatric or adult age), name of the clinician that requested the analysis, and genetic laboratory where diagnostic tests were performed. No other clinical or identifier data were collected. Approval by the independent Ethics Committee of the 12 de Octubre University Hospital (Madrid) was obtained prior to the data submission call.

Detailed information of participants (hospitals and geographical areas) is provided in the authors’ list (MITOSPAIN Working Group). Overall, 39 hospitals and diagnostic reference centres reported cases belonging to 49 of the 50 Spanish provinces.

### 2.2. Selection of Genes

For this study, we selected 293 mitochondrial genes with the following biological roles, according to [1]: (1) synthesis of complexes, assembly factors and electron carriers for oxidative phosphorylation (OXPHOS); (2) maintenance of mtDNA, including nucleoside homeostasis; (3) expression of mtDNA that affects the synthesis, processing, and modification of mitochondrial RNAs (ribosomal, transfer, and messenger) and biogenesis of mitochondrial ribosomes; (4) biogenesis of enzymatic cofactors (Fe-S clusters); (5) homeostasis and mitochondrial quality control that affect the importation of proteins into the mitochondria, lipid metabolism, mitochondrial morphology (fusion/fission, organisation of mitochondrial crests), quality control of abnormal proteins, and mechanisms of apoptosis/autophagy; and (6) OXPHOS-related energy metabolism, including defects in the Krebs cycle, pyruvate metabolism, and transport of specific metabolites.

We initially received 3274 cases that had been diagnosed with an MD between 1990 and 2020. After this first call, we removed repeated cases and mutations of unknown significance (unsolved cases) based on the guidelines of the American College of Medical Genetics (ACMG) [14]. We finally selected 2761 cases, which were classified in different ways: paediatric vs. adult cases, and mtDNA vs. nDNA mutated genes. For mtDNA mutations, only index cases were included, and not their relatives with no clinical signs. Patients with *MT-RNR1* variants were analysed separately since, for the purposes of this work, they were considered as predisposing variants to ototoxicity related to the administration of aminoglycosides. The reason is that incomplete penetrance can be a major issue for some mtDNA mutations. In particular, the mtDNA m.1555A > G mutation in the *MT-RNR1* gene could be transmitted through many generations until the beginning of the antibiotic era, and the recent use of these drugs may have triggered the current frequency of this mitochondrial disease [15].

### 2.3. Estimation of the Number of Patients

We calculated the total incidence (number of new cases per year/Spanish population) and the incidence per age group (paediatric and adult patients) according to the 95% confidence interval (CI). Incidences were expressed as cases per million inhabitants during the period 2014–2020, because during this period, the use of NGS techniques for the diagnosis of MD was implemented, and no important variations were expected. In previous years, nDNA mutations were in general scarcely detected by the Sanger sequencing approach, probably causing a selection bias. We did not calculate the estimated prevalence because data about the actual situation were difficult to establish for a significant number of patients (for example, some of them were missed during the follow-up, or they moved to other countries).

### 2.4. Genetic Diagnoses

Owing to the extended recruitment period, different genetic techniques were used for diagnosis. For mtDNA genes, point mutations, microindels, and single large-scale mtDNA deletions were studied by polymerase chain reaction and restriction fragment length polymorphism (PCR-RFLP), minisequencing, Sanger sequencing, Southern blot and real-time PCR. In recent years, whole mtDNA was sequenced by a NGS approach. For nuclear genes, Sanger sequencing was applied until 2014, when NGS was fully implemented in the most important reference centres for diagnosis of MD. Either customized gene panels, whole exome sequencing (WES), whole genome sequencing (WGS), or RNA sequencing (RNAseq) techniques were used for diagnostic purposes. The variant classification followed ACMG recommendations [14]. In most cases, mitochondrial biomarkers (such as lactate, pyruvate, alanine, urine organic acids, GDF-15 and FGF-21) were analysed during the diagnostic workflow, and mitochondrial functional studies were performed if needed.

## 3. Results

Of the 2761 patients selected, a total of 508 patients had mutations in 112 nDNA genes (384 were paediatric patients, 121 adults, and in 3 cases no age was documented), and 1105 patients carried mutations in 28 mtDNA-encoded genes (367 were paediatric patients, 716 adults, and in 22 cases age was not documented). Furthermore, 1148 cases harboured mutations in the *MT-RNR1* gene, which are associated with aminoglycoside ototoxicity and/or non-syndromic hearing loss (464 were paediatric patients, 366 adults, and in 318 cases no age was documented). The m.1555A>G variant was the mutation most frequently reported in this group (1132 cases, 98.6%). The number of cases diagnosed per year, distributed by nDNA or mtDNA genomes and by age (paediatric or adult patients), are depicted in Figure 1.

The number of reported cases for each nDNA gene and for each mtDNA mutation is presented in Figure 2 and Figure 3, respectively. All the genes screened (either nDNA or mtDNA), the biological role, and the number of cases classified as paediatric and adult patients are detailed in Appendix A. Regarding nDNA mutations, the most frequently mutated genes were *PDHA1* and *TYMP* for paediatric and adult cases, respectively. For mtDNA mutations, beyond the *MT-RNR1* m.1555A>G mutation, the most frequent mutations were m.3243A>G and single deletions for paediatric and adult patients, respectively.

The number of reported cases secondary to nuclear genome mutations significantly increased from 2014 to 2020 when compared with that in the 1990–2013 period, owing to the implementation of NGS technologies (53.1 ± 7.9 vs. 5.8 ± 5.2 cases/year, Student’s *t*-test: *p* < 0.001) (Appendix A). Before the implementation of NGS techniques as a first-line diagnostic approach, only 22.5% of the total number of patients with nDNA mutations were diagnosed (Figure 1), and only 15.8% of the genes reported (18/112) were identified. For mtDNA mutations, considering the same period (2014–2020), a slight increase in the number of diagnosed cases was also observed (54.3 ± 16.4 vs. 34.8 ± 14.5 cases/year, *p* = 0.02).

The incidence of MD in Spain from 2014 to 2020 is summarized in Table 1. We calculated the global incidence rate in this period, distributed by paediatric or adult groups. In 2020, the Spanish population was 47,332,613 inhabitants (8,789,843 paediatrics and 38,533,770 adults). After the implementation of NGS techniques in 2014, the global incidence remained stable during the study period, being greater in paediatric patients than in adults. The global incidence rate of nDNA mutations in paediatric patients and adults was 4.58 (CI: 4.05–5.12) and 0.34 (CI: 0.27–0.41), respectively, and for mtDNA mutations was 1.76 (CI: 1.42–2.09) and 1.03 (CI: 0.90–1.15) in paediatric cases and adults, respectively. The global incidence rate of *MT-RNR1* mutations was 0.34 (CI: 0.27–0.40), being 0.34 (CI: 0.20–0.49) and 0.30 (CI: 0.23–0.36) in paediatrics and adults, respectively.

## 4. Discussion

MD diagnosis requires an integrated approach comprising imaging, pathology, electrophysiology, genetics, and biochemical examinations. Recently, the diagnostic criteria for MD were revisited; nevertheless, application of this integrated approach is necessary [16]. In this study, we only considered patients with a definitive molecular diagnosis based on the presence of pathogenic mutations in either nDNA or mtDNA, impacting different mitochondrial functions [1,2,3]. One of the reasons for this approach is that some patients who were classified as having MD according to the Morava criteria and other scales were subsequently diagnosed with other non-MD. Multiple mutations in other genes were not considered at this stage but will be included in the Spanish MD registry.

mtDNA mutations were more frequently observed than mutations in nuclear genes. Our results reinforce those previously reported by other authors. The probable explanation would be that from 1990 to 2014, diagnosis of mtDNA mutations was reliable, while the identification of mutations in nuclear genes was an extremely complex task. This is confirmed by the fact that from 2014 to 2020, the global incidence rate was similar between nDNA and mtDNA MD: 1.14 (1.02–1.25) and 1.16 (1.05–1.28) cases per million inhabitants, respectively. Overall, paediatric patients were more frequently affected by MD than adults, especially in the nDNA mutation group, where the incidence was remarkably higher than that of the adults. However, for mtDNA mutations, the incidence in adults was mildly lower than that in paediatric patients, which is consistent with previous observations [4,17,18].

Beyond the *MT-RNR1* m.1555A>G mtDNA mutation, the most frequent mtDNA mutation was m.3243A>G. As was reported in an Australian Caucasian population-based study [19], the prevalence of this mutation was 236 per 100,000 individuals, thus concluding that this mutation is highly frequent and that some patients could be eluding the diagnosis, especially those in the milder adult forms. We only included index cases while excluding pedigrees in the study design, but the high frequency observed in our cohort study confirms the high frequency of the *MT-TL1* m.3243A>G mutation. Regarding nDNA mutations, before the full implementation of NGS techniques in 2014, only 18 of the total 112 genes reported were identified. The predominance of mutations in these genes is probably explained because most of these cases could be suspected by the presence of a biomarker (such as urine organic acids, respiratory chain enzyme analysis, and other enzyme activity assessment) and/or suggestive radiological and/or clinical patterns [20]. Concerning the adult group, two genes involved in mtDNA maintenance, *TYMP* and *TK2*, were the most frequent mutated genes reported (>20 cases), which is possibly due to the use of specific serum/blood metabolites as biomarkers and the existence of Spanish reference centres for the study of these particular MD as well [21,22]. Concerning *TYMP* deficiency in adults, the high frequency observed has been previously reported with the publication of a large series of patients and with the hypothesis of a relatively high prevalence in Europeans [23]. Furthermore, the other three more common mutated genes (>15 cases) in the adult group (*POLG*, *OPA1* and *TWNKL*) are also implicated in mtDNA maintenance, and muscle molecular biomarkers (multiple mtDNA deletions and mtDNA depletion) can be analysed to guide the genetic diagnosis. Lastly, we report a great number of cases harbouring *MT-RNR1* mutations associated with deafness after aminoglycoside therapy. This is consistent with two independent studies that showed a high prevalence of the *MT-RNR1* m.1555A>G mtDNA mutation in European descent population, ranging from 0.19 to 0.21% [24,25]. As previously commented, the transmission of these mutations for many generations until the beginning of the antibiotic era would explain its high frequency [15].

Interestingly, although the study was designed to include mutations causing MD in 293 genes (258 nuclear and 35 mitochondrial) [1], in our cohort we identified mutations in only 112 of the nuclear ones (Appendix A). This finding suggests that some of these genes were missed, as most patients were initially studied using customised gene panels and not using WES or other NGS approaches. Likewise, the extremely low frequency of some particular MD diseases may explain the lack of gene identification. Moreover, even in WES studies, some mutations can be missed, and further approaches such as WGS or RNAseq are required, based on recent suggestions [26].

Concerning epidemiological results, we decided to calculate the incidence per year as a different approach compared with those of previous studies in MD. In the last six years (2014–2020), the incidence was stable for both paediatric and adult patients. In the paediatric group, an average incidence of six new cases per million inhabitants was observed. This incidence is similar to that reported by Castro-Gago et al. in north-west Spain, although in that study the selection criteria was mitochondrial respiratory chain disorders, and thus data are not comparable. Regardless, we can hypothesise that our MD population has been underestimated despite the NGS-augmented diagnostic yield. We have to consider that we only included patients with pathogenic mutations in mitochondrial genes, but not those cases that fulfilled the Morava criteria without proven mutations [16]. As our diagnostic rate for MD was approximately 30–40% of cases, in the near future, when high-throughput genetic techniques can be applied as the first step in the diagnostic workflow, we believe that the incidence will increase significantly.

The main limitation of our study is that we did not have reliable data regarding the actual status of patients, mortality, and other epidemiological variables. Thus, the prevalence studies are probably inaccurate. Correspondingly, the incidence we calculated could probably be dramatically lower than in reality, because we did not screen large populations, only a cohort of candidate patients during the study period. Some factors would reinforce this statement: (i) the lack of mitochondrial DNA repair that can lead to high mutation frequency in the general population [27]; (ii) phenotypes associated with mutations in some mtDNA genes (*MT-ND2, MT-ND3, MT-ND4*, *MT-CYB*, *MT-CO1*, *MT-CO2*, *MT-CO3*, and *MT-ATP8*) are not easily recognizable as mitochondriopathies and mtDNA pathologic mutations are not looked for [28]; (iii) the pre-natal and perinatal lethality associated with several nuclear genes make diagnosis more difficult; (vi) the incomplete phenotypic presentations in heterozygous carriers for some genes is a challenge for early diagnosis [1,2,3].

## 5. Conclusions

This is the first study reporting nationwide epidemiological data for MD in Spain and the first step towards the inclusion of patients in the official Spanish MD registry in CIBERER with the final aim of sharing our data internationally. The lack of identification of a remarkable number of mitochondrial genes necessitates the systematic application of high-throughput technologies in the routine diagnosis of MD.

## Figures and Tables

**Figure 1 genes-12-01590-f001:**
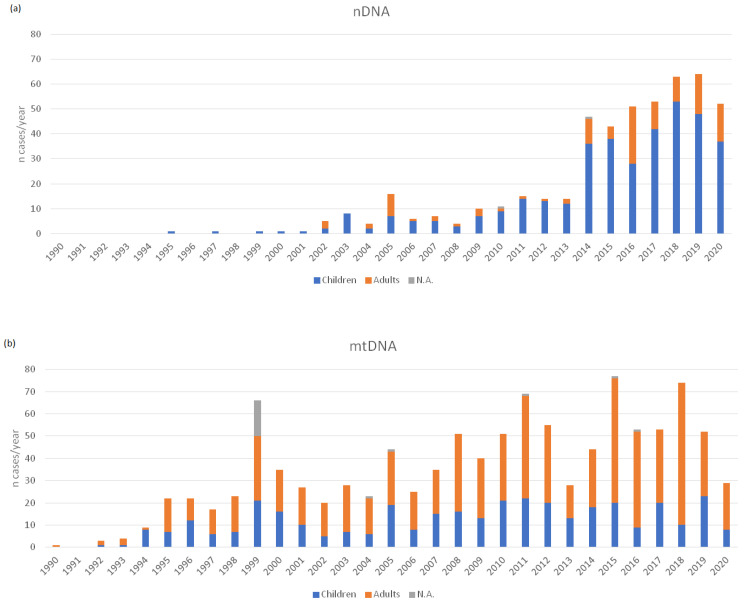
Number of cases diagnosed per year and age. (**a**) Cases harbouring nuclear DNA (nDNA) mutations; (**b**) cases harbouring mitochondrial DNA (mtDNA) mutations. Abbreviations: N.A., not available.

**Figure 2 genes-12-01590-f002:**
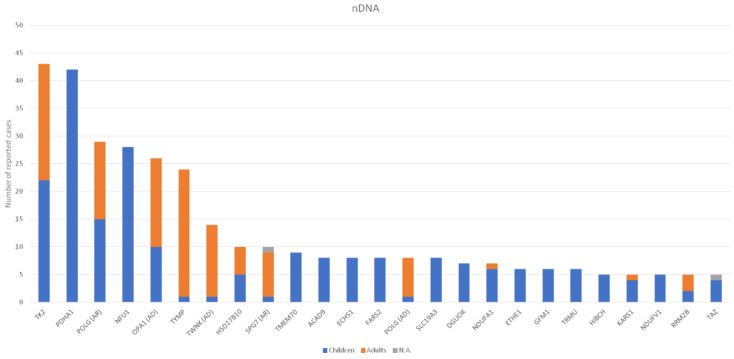
Number of reported cases for nDNA mutations distributed by age. Only genes with more than four cases are depicted. The rest of mutated genes and number of patients reported are detailed in Appendix A. Abbreviations: N.A., not available.

**Figure 3 genes-12-01590-f003:**
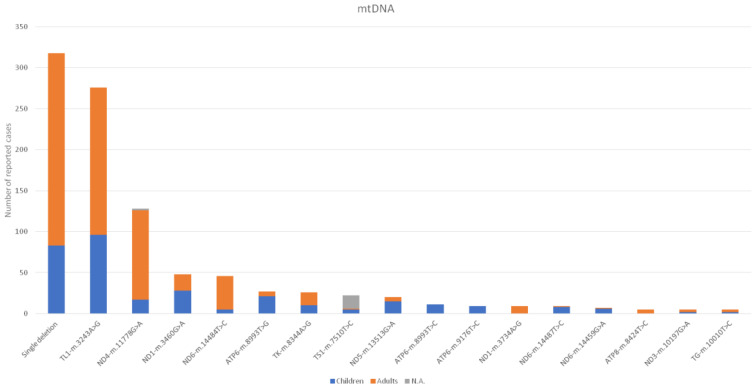
Number of reported cases for mtDNA mutations distributed by age. Only genes with more than four cases are depicted. The rest of mutated genes and number of patients reported are detailed in Appendix A. Abbreviations: N.A., not available.

**Table 1 genes-12-01590-t001:** The incidence of MD in Spain per year (from 2014 to 2020), calculated in paediatric patients, adults, and total number of patients.

Year	Paediatrics ^1^	Adults ^1^	Global ^1^
2014	6.15 (4.51–7.80)	0.95 (0.64–1.27)	1.93 (1.54–2.33)
2015	6.61 (4.91–8.31)	1.62 (1.21–2.03)	2.56 (2.10–3.02)
2016	4.21 (2.86–5.57)	1.75 (1.33–2.18)	2.22 (1.79–2.65)
2017	7.05 (5.30–8.81)	1.17 (0.82–1.51)	2.28 (1.84–2.71)
2018	7.16 (5.39–8.92)	1.95 (1.51–2.40)	2.94 (2.44–3.43)
2019	8.07 (6.19–9.94)	1.18 (0.84–1.52)	2.47 (2.02–2.92)
2020	5.11 (3.62–6.61)	0.93 (0.63–1.24)	1.71 (1.34–2.08)
2014–2020	6.34 (5.71–6.97)	1.36 (1.22–1.50)	2.30 (2.14–2.47)

^1^ Incidences are expressed as cases per million inhabitants. 95% confidence interval is indicated in brackets.

## Data Availability

Data supporting reported results can be found at Appendix A. The genes screened (either nDNA or mtDNA), their biological role and the number of cases classified as paediatric and adult patients are presented there. mtDNA mutations are detailed, while nuclear mutations are being recorded and harmonized to be presented in a near future in the RedCap platform in the context of the Spanish MD registry (https://redcap.ciberisciii.es/, accessed on 6 September 2021).

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
