# Peer review of "The Genetic Landscape of Mitochondrial Diseases in Spain: A Nationwide Call"

_genes, 2021, doi:10.3390/genes12101590_

Round 1

Reviewer 1 Report

Authors in the manuscript have reported the epidemiological data for mitochondrial disorders in Spain. They have cataloged for 3274 cases out of which frequency is reported for 2761 patients. The manuscript has been easy to comprehend, scientifically sound, and has clearly defined the merits and the limitations of the study. There are few points that authors can address in their manuscript.

  1. Have the authors included in the cases reported in reference 8 while compilation of the data for the study described in the manuscript? 2. Lines 103-105 on page 3 can be explained in little detail as to why they were analyzed separately. The rationale though mentioned needs to be explained a little more. 3. Why the incidences were only expressed during the period of 2014-2020? Is there any parameter to account for the cases prior to that? 4. It is really interesting to observe such high frequencies of mutation in the mt-rnr1 gene. It will be nice to report the mutations of nuclear and mitochondrial genes as a supplementary excel file. This will help in extending the utility of this manuscript to screen for the novel or known mutations in patients with mitochondrial diseases in the future.

Author Response

Authors in the manuscript have reported the epidemiological data for mitochondrial disorders in Spain. They have cataloged for 3274 cases out of which frequency is reported for 2761 patients. The manuscript has been easy to comprehend, scientifically sound, and has clearly defined the merits and the limitations of the study. There are few points that authors can address in their manuscript.

Thank you very much for your positive comments.

Have the authors included in the cases reported in reference 8 while compilation of the data for the study described in the manuscript?

In reference 8, the inclusion criteria for patients were different from those of our work, since the diagnosis of a mitochondrial disease was based on different biomarkers, but not solely in genetic diagnosis, as we did here. We do not know whether some of those patients were recruited for the present study, but the nationwide call included the centres that participated in the reference 8.  

Lines 103-105 on page 3 can be explained in little detail as to why they were analyzed separately. The rationale though mentioned needs to be explained a little more.  Thanks for this comment.

We have added a paragraph explaining better this issue and a reference supporting our statements: “Incomplete penetrance can be a major issue for some mtDNA mutations. In particular, the mtDNA m.1555A>G mutation in the MT-RNR1 gene has been associated with hearing loss. This mutation is deleterious when associated with aminoglycosides, it could be transmitted for many generations until the beginning of the antibiotic era, and the recent use of these drugs may have triggered the current frequency of this mitochondrial disease”. (Pacheu-Grau D, et al. Mitochondrial pharmacogenomics: barcode for antibiotic therapy. Drug Discov Today. 2010 Jan;15(1-2):33-9.)  

Why the incidences were only expressed during the period of 2014-2020? Is there any parameter to account for the cases prior to that?.

In the revised version of the manuscript, we have better explained what we meant about this issue. Since NGS techniques were not standardized prior 2014, and the possibility of a bias selection was high, especially for nuclear genes, we calculated the incidence in this latter period. In fact, the incidence during these years was stable, but it will probably increase in a near future by the routine application of WES or WGS approaches for diagnosis. 

It is really interesting to observe such high frequencies of mutation in the MT-RNR1 gene. It will be nice to report the mutations of nuclear and mitochondrial genes as a supplementary excel file. This will help in extending the utility of this manuscript to screen for the novel or known mutations in patients with mitochondrial diseases in the future.

We agree with you that this information will improve the message of the manuscript. We have now included the mtDNA mutations as a supplementary material. For nuclear mutations, we are recording them in a customized database that will be available in the near future, but in this moment, the harmonization work to express all the variants using the same nomenclature is too much time-consuming (we have some variants expressed at the protein level and other at genomic level). Moreover, other mutations are pending to be published, and we do not have the permission of our colleagues in charge of those patients to report them. Once we have the mutations uniformly recorded, they will be uploaded in our REDCap platform containing the Spanish MD registry. We have briefly described this situation in the revised version of the manuscript (supplementary material).

“Supplementary Table S2. The genes screened (either nDNA or mtDNA), their biological role as stated by Frazier et al. (reference 1) and the number of cases classified as paediatric and adult patients are presented. mtDNA mutations are detailed, while nDNA mutations are being recorded and harmonized to be presented in the near future in the REDCap platform in the context of the Spanish MD registry”.

Reviewer 2 Report

This study reports a valuable database of the identified mitochondrial diseases (MD) in Spain between 1990-2020. The MD-related genes considered were selected from previous reviews. The manuscript also comments on how changes in the sequencing techniques effected the quality of data. The manuscript does not provide information on health status or treatment outcome.

I summarized my comments in the following points.

  • The reviewer recommends using more distinguishable coloring on all figures.
  • Please provide the raw data on figures as supporting tables.
  • Fig. 2-3 the x axis labels are illegible, partly because of the image resolution partly because of letter size.
  • Fig. 2-3. Please show all considered nuclear and mitochondrial genes, even if not detected. It would be also useful to list the biological roles of all genes considered as detailed in Methods (2.2). (The reviewer suggests the usage of heatmaps.)
  • It would be intriguing to know in how many cases were multiple mutations identified and which genes were affected.
  • Lines 223-224 The manuscript discusses the limitations of this data collection method, which is contributed to the usage of gene specific panels or less advanced sequencing techniques. It would be intriguing to see the distribution of genes identified with the various techniques.

Author Response

This study reports a valuable database of the identified mitochondrial diseases (MD) in Spain between 1990-2020. The MD-related genes considered were selected from previous reviews. The manuscript also comments on how changes in the sequencing techniques effected the quality of data. The manuscript does not provide information on health status or treatment outcome. I summarized my comments in the following points.

The reviewer recommends using more distinguishable coloring on all figures. Please provide the raw data on figures as supporting tables.

We agree with you and we have modified the figures. We have also included a supporting table as supplementary material containing raw data.

Fig. 2-3 the x axis labels are illegible, partly because of the image resolution partly because of letter size.

Thanks for this suggestion. We have improved the readability of Figure 1. For Figures 2 and 3, we have reduced the number of genes represented in x-axis as those with 5 or more cases, in order to improve the quality of both figures.

Fig. 2-3. Please show all considered nuclear and mitochondrial genes, even if not detected. It would be also useful to list the biological roles of all genes considered as detailed in Methods (2.2). (The reviewer suggests the usage of heatmaps.)

We agree with you and we have now included a single file (Excel format) with 3 worksheet tables as supplementary material, containing: 1) all nuclear genes considered, their biological roles and the number of cases classified as paediatric and adult patients, 2) mtDNA genes with the same information plus the type of mutation, and 3) MT-RNR1 cases.

It would be intriguing to know in how many cases were multiple mutations identified and which genes were affected.

Thanks for this comment. However, in this call we considered only the pathogenic mutations in the best candidate gene, supported by phenotype, genotype, functional analysis when needed and other diagnostic tools. In the Spanish MD registry (supplementary material), we will have the possibility to include the genetic information suggested by you. We have added a sentence in the discussion section addressing this issue.

Lines 223-224 The manuscript discusses the limitations of this data collection method, which is contributed to the usage of gene specific panels or less advanced sequencing techniques. It would be intriguing to see the distribution of genes identified with the various techniques.

We completely agree with you and we tried to add this variable in the initial design of the work. However, it was not possible to collect such information for a remarkable number of cases. The explanation for this is that the study was multicentric (40 centres) and it covered a long recruitment period (30 years) and, thus, in many hospitals this information was missing.

Reviewer 3 Report

                This paper reports the results of a survey of mitochondrial diseases reported in the Spanish population between 1990 and 2020.  The data includes the number of patients categorized by pediatric or adult and nuclear or mtDNA mutation. The authors statistically analyzed frequency of case reporting and incidence of mitochondrial disease in the Spanish population, but did not report their basic data set, i.e., identification of genes and mutation location, function of the mutated gene, gene location [nuclear or mtDNA], etc.  I think most readers of this paper would be interested in the specific genes involved in mitochondrial diseases in this population.  The absence of this information is a weakness of the paper.  Part of this information is presented in Figures 2 and 3, but these figures are almost completely unreadable.  I strongly suggest that these figures be converted to Tables which would allow additional information to be displayed in a more readable form.  If these Tables are too large to be included in the paper, they can be included in the Supplementary Materials.

                Since a significant number of the cases involved MT-RNR1, this group should be better characterized.  It should be introduced in the Introduction by defining MT-RNR1 and discussing its significance and why it is categorized separately from nuclear mutations and other mtDNA mutations.  MT-RNR1 was mentioned in the Discussion (line 217 -220), but without context it will be difficult for most readers to judge the significance of the report.

                In the Discussion both the higher frequency of mtDNA mutations relative to nuclear mutations (line 184 and following) and the fact that many of the known mutations causing mitochondrial diseases were not observed in this survey (line 221 and following) were discussed.  In each case, limits of the statistical analysis were discussed (e.g., small cohort sizes [line 246] and unreliability of the reported data [line 242], etc.), but other explanations such as lack of mitochondrial DNA repair (higher mutation frequency), pre-natal lethality, effects of heterozygosity, etc., were not discussed. I think that this discussion could be expanded.

                Also, it was stated that only 114 of the expected 289 genes had been identified (line 222).  In Supplementary Materials about 450 candidate nuclear genes are listed.  Where does the 289 number come from?

                In sum, I find this to be an interesting paper and I do not see any major scientific problems, however, I think more specific presentation of the survey data and expanded background and discussion could make this paper of interest to a broader group of readers.

Author Response

This paper reports the results of a survey of mitochondrial diseases reported in the Spanish population between 1990 and 2020.  The data includes the number of patients categorized by pediatric or adult and nuclear or mtDNA mutation. The authors statistically analyzed frequency of case reporting and incidence of mitochondrial disease in the Spanish population, but did not report their basic data set, i.e., identification of genes and mutation location, function of the mutated gene, gene location [nuclear or mtDNA], etc.  I think most readers of this paper would be interested in the specific genes involved in mitochondrial diseases in this population.  The absence of this information is a weakness of the paper.  Part of this information is presented in Figures 2 and 3, but these figures are almost completely unreadable.  I strongly suggest that these figures be converted to Tables which would allow additional information to be displayed in a more readable form.  If these Tables are too large to be included in the paper, they can be included in the Supplementary Materials.

We agree with you and we have modified the figures and we have also included a supporting table as supplementary material containing raw data. Concerning the figures, we have improved the readability of all figures, as also suggested by reviewer 2. Moreover, we have now included a single file (Excel format) with 3 worksheet tables as supplementary material, containing: 1) all nuclear genes considered, their biological roles and the number of cases classified as paediatric and adult patients, 2) mtDNA genes with the same information plus the type of mutation, and 3) MT-RNR1 cases.

Concerning the nuclear mutation description, we are recording them in a customized database that will be available in the near future, but in this moment, the harmonization work to express all the variants using the same nomenclature is too much time-consuming (we have some variants expressed at the protein level and other at genomic level). Moreover, other mutations are pending to be published, and we do not have the permission of our colleagues in charge of those patients to report them. Once we have the mutations uniformly recorded, they will be uploaded in our REDCap platform containing the Spanish MD registry. We have briefly described this situation in the revised version of the manuscript (supplementary material).

“Supplementary Table S2. The genes screened (either nDNA or mtDNA), their biological role as stated by Frazier et al. (reference 1) and the number of cases classified as paediatric and adult patients are presented. mtDNA mutations are detailed, while nDNA mutations are being recorded and harmonized to be presented in the near future in the REDCap platform in the context of the Spanish MD registry”.

Since a significant number of the cases involved MT-RNR1, this group should be better characterized.  It should be introduced in the Introduction by defining MT-RNR1 and discussing its significance and why it is categorized separately from nuclear mutations and other mtDNA mutations.  MT-RNR1 was mentioned in the Discussion (line 217 -220), but without context it will be difficult for most readers to judge the significance of the report.

We agree with you and we have added this information in the revised version of the manuscript: “Incomplete penetrance can be a major issue for some mtDNA mutations. In particular, the mtDNA m.1555A>G mutation in the MT-RNR1 gene has been associated with hearing loss. This mutation is deleterious when associated with aminoglycosides, it could be transmitted for many generations until the beginning of the antibiotic era, and the recent use of these drugs may have triggered the current frequency of this mitochondrial disease”. (Pacheu-Grau D, et al. Mitochondrial pharmacogenomics: barcode for antibiotic therapy. Drug Discov Today. 2010 Jan;15(1-2):33-9.)                  

 In the Discussion both the higher frequency of mtDNA mutations relative to nuclear mutations (line 184 and following) and the fact that many of the known mutations causing mitochondrial diseases were not observed in this survey (line 221 and following) were discussed.  In each case, limits of the statistical analysis were discussed (e.g., small cohort sizes [line 246] and unreliability of the reported data [line 242], etc.), but other explanations such as lack of mitochondrial DNA repair (higher mutation frequency), pre-natal lethality, effects of heterozygosity, etc., were not discussed. I think that this discussion could be expanded.

Thank you for the suggestion. We agree with you and we have added these interesting aspects in the discussion together with some references supporting this information: “Some factors would reinforce this statement. i) The lack of mitochondrial DNA repair that can lead to high mutation frequency in the general population [27]. ii) Phenotypes associated to mutations in some mtDNA genes MT-ND2-4, MT-CYB, MT-CO1-3, and MT-ATP8 are not easily recognizable as mitochondriopathies and mtDNA pathologic mutations are not looked for [28]. iii) the pre-natal and perinatal lethality associated with several nuclear genes difficult the diagnosis. vi) the incomplete phenotypic presentations in heterozygous carriers for some genes is a challenge for early diagnosis [1-3]”.  

Also, it was stated that only 114 of the expected 289 genes had been identified (line 222).  In Supplementary Materials about 450 candidate nuclear genes are listed.  Where does the 289 number come from?.

You are right and thanks for the comment. In the manuscript, we finally looked for the 293 genes stated in reference 1 (plus 4 more that were recently discovered). However, the list of genes included in the supplementary material are those that will be included in the Spanish MD Registry. For this registry, the list of genes is longer since we included mitochondrial genes with no proven mutations yet (e.g., COQ10A, COQ10B, and others), genes recently described, an also mitochondrial genes that are not currently considered in the recent revisions as primary mitochondrial diseases (e.g., fatty acid oxidation defects). Since the creation of this registry has been a great effort, we prefer to do it as extensive as possible and after that, data curation will allow to select specific genes. We have clarified this aspect in the supplementary material legend: “In the Spanish register, the list of genes is longer than that stated in the present manuscript since it also includes mitochondrial genes not involved yet in diseases of other nuclear genes affecting mitochondrial pathways such as free fatty oxidation defects.”

After carefully reassessing our data, we have slightly modified the number of nuclear genes reported, from 114 to 112. The reason of this discrepancy is that in the first version we had considered two disorders that can be autosomal recessive and dominant (e.g., mutations in POLG) as different genes. Sorry for this mistake. 

                In sum, I find this to be an interesting paper and I do not see any major scientific problems, however, I think more specific presentation of the survey data and expanded background and discussion could make this paper of interest to a broader group of readers.

Thank you very much for this positive conclusion.

Round 2

Reviewer 2 Report

The paper went though a careful editing after the first review. I would emphasize that Supplementary Table S2 - that lists the genes screened, the biological role and the number of cases classified - made the paper more accessible for the scientific community. The quality of the figures and legends improved since the first submission. The discussion of the results become more clear. The reviewer understands that the long recruitment period of this study did not allow to address some of the intriguing questions that would enable deeper analysis of the correlations between genetic factors leading to mitochondrial diseases. The reviewer recognizes the importance of the work, and encourages the authors to continue the data collection on genetic mutations together with the health status and treatment outcome.